# Characterization of p53 p.T253I as a pathogenic mutation underlying Li-Fraumeni Syndrome

**Nathaniel C. Holcomb[1], Amanda M. Harrington[1,2], Hong Pu[1], Berina Halilovic[1], Nathan R. Shelman[1,3], Shulin Zhang[1,3], Catherine Sears[1,3], Terra Armstrong[1], Brent Shelton[1,4], Lauren Corum[1,4], John A. D'Orazio [1,2]***

1 The Markey Cancer Center, University of Kentucky College of Medicine, Lexington, Kentucky, United States of America, 2 Department of Pediatrics, University of Kentucky College of Medicine, Lexington, Kentucky, United States of America, 3 Department of Pathology, University of Kentucky College of Medicine, Lexington, Kentucky, United States of America, 4 Department of Cancer Biostatistics, University of Kentucky College of Medicine, Lexington, Kentucky, United States of America

\* jdorazio@uky.edu

## Abstract

We identified a germline *TP53* c.758C > T (p.T253I) mutation in the *TP53* tumor suppressor gene in a pediatric adrenocortical carcinoma (ACC) patient. Characteristic of pathogenic p53 mutations, we observed upregulation of total p53 protein levels in the patient's ACC and concurrent suppression of the wild-type (WT) *TP53* allele. As ACC can be associated with Li-Fraumeni Syndrome (LFS) and the mutation has not yet been linked to LFS, we sought to characterize the functionality of the T253I mutation. We acquired p53$^{-/-}$ HEK293 cells and stably transduced them with GFP-tagged wild type (T253) or T253I p53 as well as two established pathogenic p53 mutants (C176Y and R213X). Compared to p53 WT, levels of T253I p53 increased while MDM2 levels decreased, suggesting a loss of MDM2-mediated regulation of T253I p53. Additionally, T253I showed a reduction in DNA damage responsive events, diminished DNA binding capabilities, and blunted transactivation capacity. These experimental data lead us to conclude that T253I represents a pathologic variant in *TP53* that may predispose to LFS-associated tumors.

## Introduction

Li-Fraumeni Syndrome (LFS) is a cancer predisposition syndrome caused by an inherited defect in one allele of the *TP53* tumor suppressor gene or similar loss of function of a gene in the p53 damage response pathway [1,2]. LFS significantly raises lifetime risk of a variety of pediatric and adult-type cancers including sarcomas, brain tumors, melanoma, breast cancer, adrenocortical carcinoma (ACC) and leukemia [3,4]. Penetrance of LFS is variable, but it is estimated that germline loss-of-function mutations in *TP53* can be associated with up to a 70% lifetime risk of cancer in men and up to a 90% lifetime risk of cancer in women, with a large number of

**Data availability statement:** We have made all data used in this manuscript publicly available at the online data repository FigShare with the following DOI: 10.6084/m9.figshare.27902106. The complete data set for each figure in the manuscript is available at the following link https://figshare.com/articles/dataset/Figure_1_A_and_B_-_Data_and_Analysis/27902106 You can also find this data set by searching the DOI listed above on DOI.org.

**Funding:** Kentucky Pediatric Cancer Research Trust Fund NCI P30 CA177558 NIH P20 GM121327.

**Competing interests:** The authors have declared that no competing interests exist.

patients presenting with their first malignancy in childhood, adolescence and young adulthood [5,6]. It is now appreciated that LFS represents a spectrum of clinical phenotypic severity based on specific *TP53* mutations involved [7,8]. The benefits of clinical surveillance of LFS patients with known pathogenic mutations in *TP53* are well established [9,10], therefore it is important to identify patients with LFS as rapidly as possible to offer appropriate cancer surveillance and prophylaxis management as well as to extend genetic testing to at-risk family members.

Here we describe the function of a p53 variant, c.758C > T (T253I), observed in an infant who presented with ACC and a germline variant of uncertain significance (VUS) in TP53. While pediatric ACC is associated with LFS [11,12], and the p53 T253I mutation has been identified as a temperature sensitive mutation in yeast [13], there have been no additional published studies focusing on the functional impact of the T253I mutation on p53 in mammalian cells. By comparing the function of T253I *in vitro* to both p53 WT and two established loss of function p53 mutations also identified in pediatric patients (C176Y and R213X [14,15]), our findings suggest that T253I represents a pathogenic loss-of-function mutation in p53 that may be associated with LFS.

## Materials and methods

### Cells and cell culture

HEK *TP53* wt/wt (ATCC Cat. #CRL-1573), HEK Crispr-deleted *TP53*-null (*TP53*−/−; Ubigene Cat. #YKO-H177), and all stably transduced derivatives of HEK *TP53*−/− cells were maintained in DMEM (Thermo Scientific) supplemented with 10% FBS (Gemini Bio-Products) in a 37°C incubator with 5% $CO_2$ and maintained in log growth phase.

### Stable expression mutant p53 genes

HEK p53−/− cells were transduced with a lentivirus containing GFP-tagged p53 (p53 WT-GFP) or mutant derivatives of p53: C176Y-GFP, R213X-GFP or T253I-GFP, all under a cytomegalovirus (CMV) promoter (Origene Cat. # RC200003L4V). Stable lines were selected (puromycin) and expression of constructs confirmed by GFP fluorescence and immunoblotting.

### Cell growth rate and GFP expression detection by flow cytometry

Doubling times for HEK p53−/− and HEK p53−/− cells complemented with p53 WT-GFP, C176Y-GFP, R213X-GFP, or T253I-GFP were tested by CellTrace Violet proliferation assay (Thermo Fisher Cat. #C34571) as described [16].

### Subcellular fractionation and immunoblotting

HEK p53−/− and HEK p53−/− cells complemented with p53 WT-GFP, C176Y-GFP, R213X-GFP, or T253I-GFP were grown in log phase. Nuclear and cytoplasmic fractions were isolated using the NE-PER Nuclear and Cytoplasmic Extraction Kit (Thermo Scientific Cat. #78833) and immunoblotted for the indicated proteins using GAPDH and histone H3 as purity controls. For whole cell lysate experiments,

immunoblotting was performed according to manufacturer's instructions and published methods [17,18]. A full list of antibodies used is available in S1 Table.

## RT-PCR analysis

Gene expression measurement for p21 and MDM2 was conducted using real-time polymerase chain reaction (rtPCR) analysis as previously described [18] under the following conditions: 25°C for 10 minutes, 42°C for 60 minutes, and 95°C for 5 minutes. Primer pairs and TaqMan probes (Thermo Scientific. Cat. # 4331182) were used to determine p21 (Cdkn1a Gene Expression Assay ID Hs00355782_ml) and MDM2 (Gene Expression Assay ID: Hs99999008_m1) gene expression levels. Gene expression data were normalized to 18s gene expression of eukaryotic 18S rRNA endogenous control (Thermo Scientific Cat. # 4319413E).

## DNA binding assay

p53 DNA binding capacity was measured using a colorimetric p53 Transcription Factor ELISA Assay Kit (Abcam Cat. #ab207225) as previously published [19]. Signal strength was calculated for each p53 condition with or without damage, normalized using p53$^{-/-}$ cells as a negative control and compared relative to undamaged WT-GFP-complemented p53$^{-/-}$ cells (positive control).

## Transcriptional activity assay

p53 transcriptional activity was measured using the Dual-Glo luciferase assay system (Promega Cat#E2940) as previously published [20] with two plasmids, luc2P/P53 response element (Promega Cat #E393A) and hRluc/CMV (Promega Cat. #E365A) using log-phase HEK p53$^{-/-}$ and p53$^{-/-}$+WT-GFP, C176Y-GFP, R213X-GFP, or T253I-GFP cells. Data reflect three experimental replicates, each with eight technical replicates for each cell line.

## CHIP-PCR assay

The ability of p53 to bind *in vitro* to DNA was measured using chromatin immunoprecipitation and PCR amplification. Chromatin immunoprecipitation was performed using the SimpleChIP® Enzymatic Chromatin IP Kit (CST Cat #9003). Briefly, genomic DNA from whole cell extracts was crosslinked to all proteins actively binding DNA, fragmented into small (150–900 bp) lengths, then immunoprecipitated with p53 antibody or a positive control with H3 antibody (CST Cat #4620). After purification of the DNA fragments, PCR amplification using ChIP suitable p21 primers was performed p21 (FOR: CCC ACA GCA GAG GAG AAA GAA. REV: CTG GAA ATC TCT GCC CAG ACA. Sigma Aldrich Cat# 17–613), normalized to the input expression of RPL30 (Exon 3 Primers from Sigma Aldrich Cat# 7014S), for each p53 condition. Data reflect the mean and standard deviation of three experimental replicates.

## Statistics

Statistical analyses were performed in conjunction with our institutional Biostatistics and Bioinformatics Shared Resource Facility. In all figures, data are presented as mean±standard error of proportions (percents). One-Way ANOVA for multiple group comparisons was used to assess statistical significance of differences amongst and between groups of interest. An experiment-wise significance level of 5% was set prior to data analysis and appropriate post-hoc testing (Newman-Keuls Multiple Comparison Test) was employed. Where appropriate, statistical significance is denoted by symbols (* indicates $p<0.05$ when comparing the sample to p53 -/-, † indicates $p<0.05$ when comparing the sample to p53 WT). GraphPad Prism V 5.01 was used to perform the statistical analyses presented in this manuscript with the exception of the growth rates, which were determined by exponential decay regression analysis performed by Sigmaplot V 15.0. In this case, the amount of signal remaining at each timepoint was calculated as a percentage relative to the initial signal at time 0 and the

half-life was found by solving the exponential decay regression curve y = a * exp(-b * x) for 50% signal remaining. This half-life value was then used to represent the population doubling time.

### Ethical statement

This research was developed from a clinical study called "Project Inherited Cancer Risk" offered at the University of Kentucky to provide germline sequencing for a panel of cancer-associated genes. The study was vetted and approved by our Institutional Review Board (IRB), and permission for the use of the patients' data for research purposes was explicitly granted by the consenting legal guardian.

## Results

### Detection of p53 T253I germline mutation in ACC patient

A pediatric patient was recently diagnosed with a stage 1 adrenocortical carcinoma. Tumor molecular profiling identified a c.758C>T variant in the *TP53* gene (variant allele frequency 72%) resulting in a threonine to isoleucine substitution at position 253 of p53 protein. The resected ACC tumor tissue stained more strongly for total p53 protein levels than adjacent non-neoplastic tissue, and the staining was strongly nuclear (S1 Fig), similar to tumor histologies of p53-mutant tumors in previously published studies [21]. ACC is a tumor characteristic of LFS, a cancer predisposition syndrome caused by germline mutations that disrupt p53 tumor suppressor function [1,2]. Indeed, 27 of 39 pediatric ACC patients had mutations in *TP53*, an incidence rate of almost 70% in a hallmark study of pediatric cancers [12]. The *TP53* c.758C>T variant was also found in the patient's germline, however since we observed that the *TP53* c.758C>T *T253I* variant is classified as a variant of uncertain significance (VUS) in the ClinVar database, we therefore sought to characterize the impact of the T253I mutation on p53 function.

### Stable expression of clinically observed p53 mutants

We hypothesized that the p53 T253I mutation may result in a loss of function in the tumor suppressor p53 since pediatric ACC can be associated with LFS [22,23]. To test this, we developed *TP53* CRISPR-deleted HEK293 cells (HEK p53$^{-/-}$) as a system in which p53-null cells could be complemented with wild type (WT) or mutant p53 to compare cell growth and p53-mediated damage responses. We observed no p53 expression in HEK p53$^{-/-}$ cells in contrast to unmanipulated HEK293 p53$^{+/+}$ cells or WT-complemented p53$^{-/-}$ HEK293 cells (S2 Fig). We found that the GFP-tagged p53 WT band was of higher molecular weight, about 75 kDa compared to 53 kDa, than native p53 WT, likely because of the GFP tag attached to p53 constructs in the complemented condition. Further, we noted that expression of WT-GFP p53 was considerably stronger than native p53 present in parental *TP53*-intact HEK293 cells (6.1-fold, comparing lanes 1 and 3, $p < 0.05$), presumably because expression of the construct was under the control of a CMV promotor.

We next developed a T253I *TP53* construct as well as two known loss-of-function p53 mutants for use as experimental controls – C176Y [24,25] and R213X [26,27] (S3 Fig). Each GFP-tagged *TP53* construct was independently and stably transduced into *TP53*-null HEK293 cells by lentiviral-mediated delivery. We took advantage of the GFP-tags on each of the complemented constructs to verify expression and to measure p53 construct expression through cellular green fluorescence by flow cytometry (Fig 1). Uncomplemented p53$^{-/-}$ cells had negligible levels of fluorescence at 488 nm, while HEK293 cells complemented with WT-GFP, C176Y-GFP, R213X-GFP and T253I-GFP p53 constructs each exhibited stronger expression of green fluorescence (Fig 1a). We observed that the expression of each of the p53 mutants tested (C176Y-GFP, R213X-GFP, T253I-GFP) was significantly higher than that of the wild type p53-GFP complemented condition, with median fold induction values of 3.0, 2.6 and 4.1 ($p < 0.05$) compared to p53 T253-GFP (WT), respectively (Fig 1b). To confirm that these levels of p53 were not detrimental to normal cellular growth, we performed a dye dilution assay to measure cell doubling times using the Cell Trace Violet Proliferation Kit (Thermo Scientific). Population doubling time

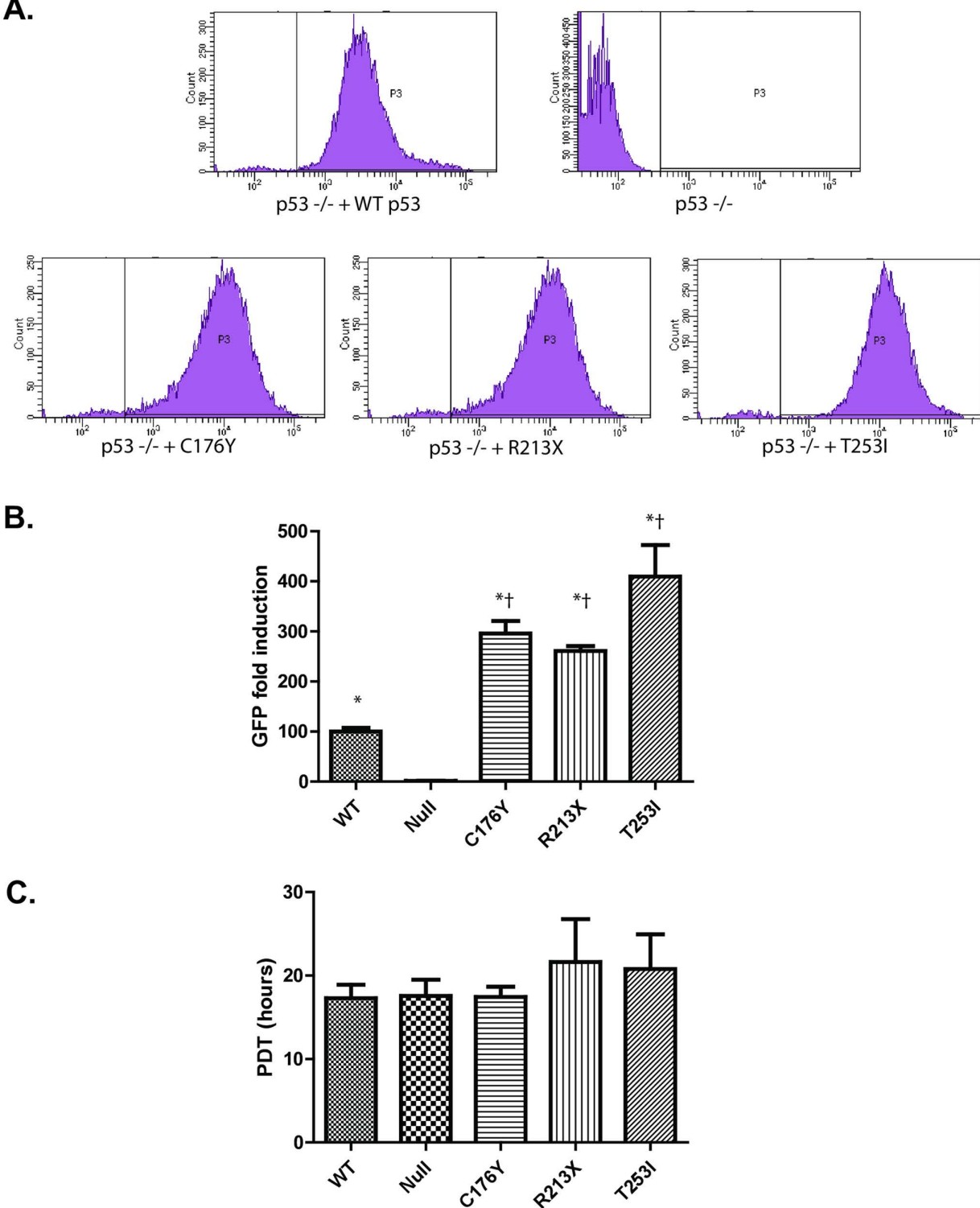

**Fig 1. Stable incorporation of p53 lentiplasmids is detectible and does not impact cellular growth rates. (A)** HEK293 p53$^{-/-}$ cells were stably complemented either WT-GFP, C176Y-GFP, R213X-GFP or T253I-GFP p53. After selection, cells were assayed for GFP fluorescence by flow cytometry. Shown are representative histograms (green fluorescence) of the geometric means of each complemented cell type. **(B)** Graphical representation of fluorescent intensity of each p53 cell line. Median GFP fluorescent intensity was calculated for each cell line and normalized to the p53 WT expression

levels. Data represent the mean of three experimental replicates. **(C)** Cell growth kinetics of stably complemented HEK293 p53$^{-/-}$ cells were measured by use of a CellTrace Violet assay. After incorporating a fluorescent dye, cells were harvested 24, 48, 72, and 96 hours after treatment, fixed, and dye intensity was measured by flow cytometry. Geometric mean intensity at each timepoint was used to calculate an exponential decay regression curve and ultimately a population doubling time (PDT). Data represent the mean PDT of three experimental replicates and significance is denoted by * or † (* indicates $p < 0.05$ when comparing the sample to p53 -/-, † $p < 0.05$ when comparing the sample to p53 WT).

was not statistically different ($p = .7903$, 1-Way ANOVA) between HEK p53$^{-/-}$ cells and p53$^{-/-}$ cells complemented with either p53 WT-GFP, C176Y-GFP, R213X-GFP, or T253I-GFP (Fig 1c). Therefore, we concluded that expression of any of the p53 constructs did not impact cell proliferation.

**T253I shows abnormal autoregulatory feedback with MDM2**

Given the higher levels of GFP signal in the mutant p53 expressing cells, we measured total p53 protein levels by immunoblotting in p53$^{-/-}$ HEK293 cells and in those complemented with WT-GFP, C176Y-GFP, R213X-GFP, or T253I-GFP p53. Whereas no p53 protein was found in in p53$^{-/-}$ HEK293 cells, there was evidence of p53 expression in each of the complemented cell lines. Similar to findings with GFP fluorescence by flow cytometry, we observed that levels of expression of C176Y-GFP, R213X-GFP, or T253I-GFP p53 were higher than that of WT-GFP with densitometry values 4.5-, 1.8- and 3.2-fold higher than p53 WT-GFP p53, respectively (S4 Fig).

We then considered why protein levels of p53 T253I-GFP, like those of C176Y-GFP and R213X-GFP, were stronger than p53 WT-GFP levels despite each being expressed under the same (CMV) promoter. Knowing that p53 levels are regulated mainly by an auto-feedback mechanism wherein p53 transcriptionally induces expression of MDM2, an E3 ubiquitin ligase that targets p53 for proteasomal degradation [28,29], we surmised that overexpression of mutant p53 proteins may be due to lower MDM2 levels, presumably because of loss of p53 transcriptional activity. Indeed, we observed that MDM2 levels were substantially lower in the p53$^{-/-}$ null (12.5%) and C176Y-GFP, R213X-GFP, or T253I-GFP p53 cells (15.3%, 19.5%, and 34% for p53 C176Y-GFP, R213X-GFP, and T253I-GFP, respectfully compared to p53 WT-GFP complemented cells; S4 Fig). We interpret these data to indicate that the high p53 expression levels of the mutant p53 proteins may be a consequence of insufficient MDM2 levels in the setting of defective p53 transcriptional activity, interfering with the autoregulatory feedback loop's regulation of cellular p53 levels.

Since p53 nuclear localization is critical to its function and is one of the targets for MDM2 negative regulation, we next investigated the cytoplasmic and nuclear distribution of p53 and MDM2 in p53$^{-/-}$ null, WT-GFP, or C176Y-GFP, R213X-GFP, or T253I-GFP complemented HEK293 cells. We observed that each of the p53 constructs could be detected in both nuclear and cytoplasmic fractions (Fig 2a) with cytoplasmic levels (Fig 2b, **left**) of each of the mutants higher than that of WT (fold increases of 2.6, 2.2 and 2.7 respectively). In contrast, nuclear levels (Fig 2b, **right**) of each of the mutants were much more similar to that of WT p53 (fold changes of 1.4, 1.0, and 1.5 respectively). With respect to MDM2 localization, cytoplasmic MDM2 levels were similar across all conditions examined (Fig 2c, **left**), while nuclear MDM2 levels were higher in the WT-complemented condition (Fig 2c, **right**). Indeed, each mutant line had nuclear MDM2 levels comparable to uncomplemented p53$^{-/-}$ cells, suggesting p53-mediated dysfunctional MDM2 localization in the setting of any of the mutants tested (C176Y, R213X, or T253I). We conclude that reduced nuclear MDM2 levels in p53 T253I-GFP complemented cells, similar to the p53-null or loss-of-function p53 control conditions (C176Y-GFP, R213X-GFP), may contribute to the increased total p53 expression levels by an impaired p53-MDM2 autoregulatory loop. Further, these findings suggest that the T253I mutation may impede p53 transactivation ability and MDM2 nuclear localization.

**Canonical p53 DNA damage responses are negatively impacted in p53 T253I cells**

To test the impact of the T253I mutation on p53 protein function, we measured p53-dependent and damage-mediated induction of MDM2 and p21 [30] by incubating HEK293 p53$^{-/-}$ cells complemented with WT-GFP- or C176Y-GFP,

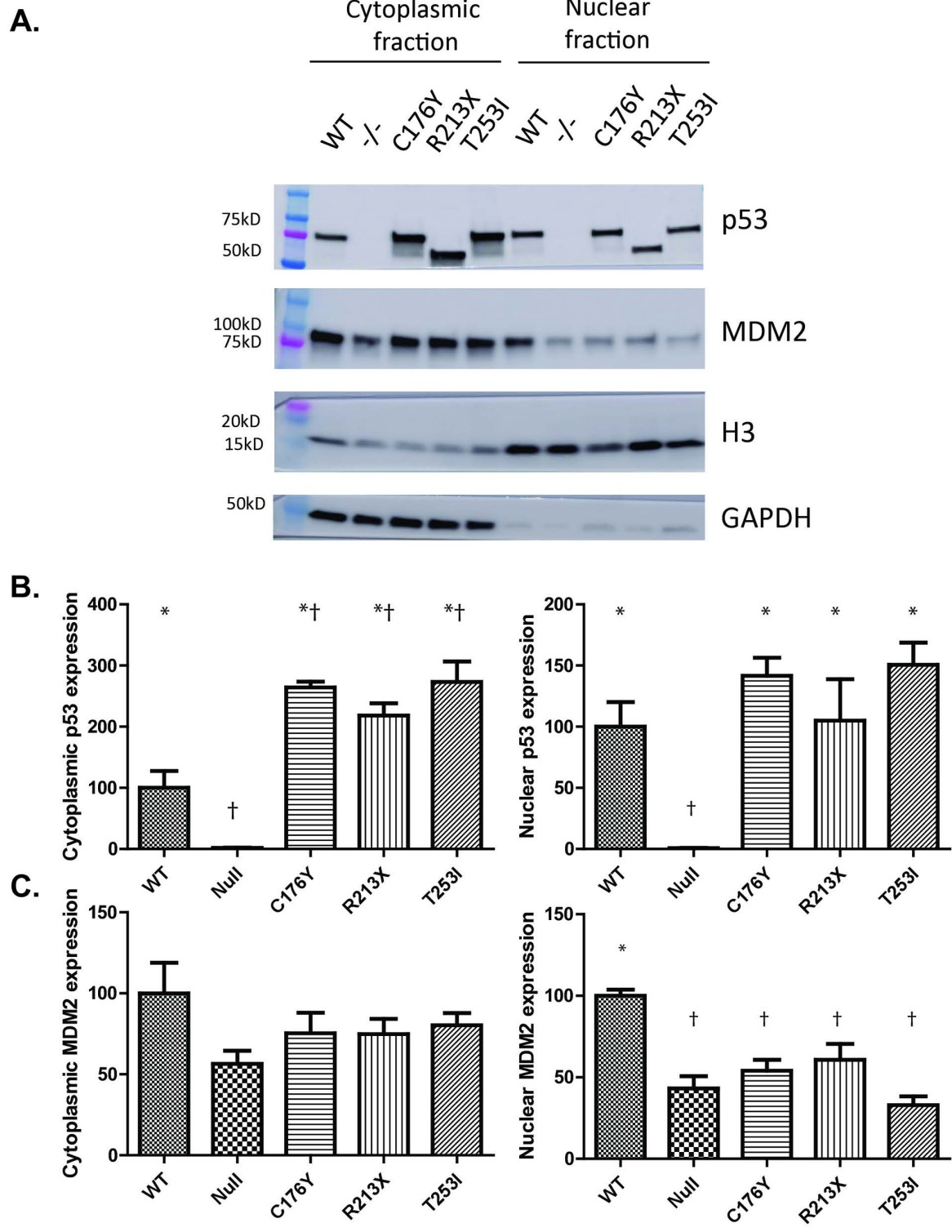

**Fig 2. Subcellular fractionation of HEK293 p53 wt and mutant proteins.** HEK293 p53⁻/⁻ cells and those complemented with WT-GFP, C176Y-GFP, R213X-GFP, or T253I-GFP p53 were subjected to subcellular fractionation to look at the distribution of p53 and MDM2 in the cytoplasm (**A, left side**) and nucleus (**A, right side**). **(B)** Graphical depictions of the cytoplasmic (**left**) and nuclear (**right**) expression levels of p53. **(C)** Graphical depictions of the cytoplasmic (**left**) and nuclear (**right**) expression levels of MDM2. In both cases, expression levels of cytoplasmic fractions were normalized to

GAPDH while expression levels of nuclear fractions were normalized to Histone 3 (H3). Data represent the mean expression levels of three experimental replicates and significance is denoted by * or † (* indicates p < 0.05 when comparing the sample to p53 -/-, † p < 0.05 when comparing the sample to p53 WT).

R213X-GFP or T253I-GFP p53 in the presence or absence of sublethal cisplatin. Kinetic studies in p53 WT-GFP cells documented significant accumulation of p21 and MDM2 protein by 2h after cisplatin exposure (S5 Fig), therefore we chose the 2h time point to characterize the impact of each of the mutations on p53 function. We observed that levels of total p53 were higher across all mutations tested compared to WT and that cisplatin did not affect total p53 levels (Fig 3a, 3b). Damage-induced up-regulation of MDM2 and p21 signals was observed in WT-GFP-complemented p53$^{-/-}$ cells, but not in p53$^{-/-}$ or C176Y-GFP, R213X-GFP or T253I-GFP-complemented p53$^{-/-}$ cells HEK293 cells (Fig 3a, 3c, 3d). Additionally, MDM2 and p21 levels in undamaged WT-GFP complemented p53$^{-/-}$ cells were significantly stronger than in the p53 mutant or null cell lines. Together these observations suggest that the T253I mutation, like known loss-of-function C176Y and R213X mutations, interferes with canonical p53 damage responses.

Given that p53 transcriptionally regulates p21 and MDM2 expression [30], we compared the ability of the T253I mutant to WT, C176Y and R213X p53 to modulate expression of p21 and MDM2 mRNA by real time PCR. We observed higher baseline of p21 (p < 0.05) and MDM2 (higher, but not significant) mRNA expression in WT-GFP-complemented p53$^{-/-}$ cells as compared to C176Y-GFP, R213X-GFP or T253I-GFP-complemented p53$^{-/-}$ cells HEK293 cells (Fig 4a, 4b). With damage, both p21 and MDM2 mRNA expression levels increase, but not significantly. These data suggest that the T253I mutation, like C176Y and R213X known loss-of-function variants, impairs p53 transcriptional activity.

Next, we compared the ability of the p53 constructs to bind a consensus p53 DNA binding sequence by ELISA. P53 WT-GFP binds to this DNA much more strongly than any of the other mutants tested (58.69%, 49.90%, and 61.43% lower for C176Y-GFP, R213X-GFP or T253I-GFP respectively; Fig 5a). In addition, sublethal cisplatin exposure increased binding of WT p53 by 92.6% (Fig 5a, **columns 1 and** 2) but did not increase binding of any of the mutants tested, suggesting T253I has impaired basal- and DNA damage-induced DNA binding capabilities. We also performed a CHIP-seq assay to measure the ability of p53 constructs to bind genomic DNA at the p21 promoter (Fig 5b). Whereas HEK p53 $^{-/-}$ cells possessed no binding capabilities, those complemented with p53-WT showed strong binding signals. In contrast, all p53 mutant cell lines (C176Y, R213X, and T253I) had significantly reduced p53 binding capability (68–80% reduction in DNA binding compared to WT). In order to directly compare transcriptional regulatory capacity of T253I with WT p53, we employed a dual luciferase reporter assay consisting of a p53-dependent promotor driving Firefly luciferase to measure p53-dependent responses, and a CMV promotor-driven *Renilla* luciferase to control for transfection. Whereas strong p53-dependent firefly luciferase signal was observed from p53-null HEK293 cells complemented with WT-GFP, there was minimal signal from either uncomplemented p53$^{-/-}$ cells or from p53$^{-/-}$ cells complemented with either C176Y-GFP, R213X-GFP or T253I-GFP (S6 Fig). Since all cell conditions had strong signal from the CMV-promoter driven *Renilla* luciferase, we concluded that observed differences were not caused by differential transfection efficiency in the assay. These data indicate that WT-complemented p53$^{-/-}$ cells alone had transcriptionally active p53 protein in this assay (Fig 5c). We conclude that although the T253I mutant p53 protein (like the C176Y or R213X mutant p53 proteins) may bind DNA in a limited capacity, its transcriptional activity is markedly impaired.

## Discussion

Here we report evidence supporting the designation of p53 T253I as a LFS-associated loss of function mutation. Observed in a young child with an ACC and germline c.758C>T *TP53* variant, the resulting T253I mutant p53 protein exhibited attenuated p53 baseline and damage-induced cell responses. The strong clinical association between pediatric ACC and Li-Fraumeni Syndrome (LFS) [31], coupled with the paradoxical overexpression of p53 in the tumor, led us to

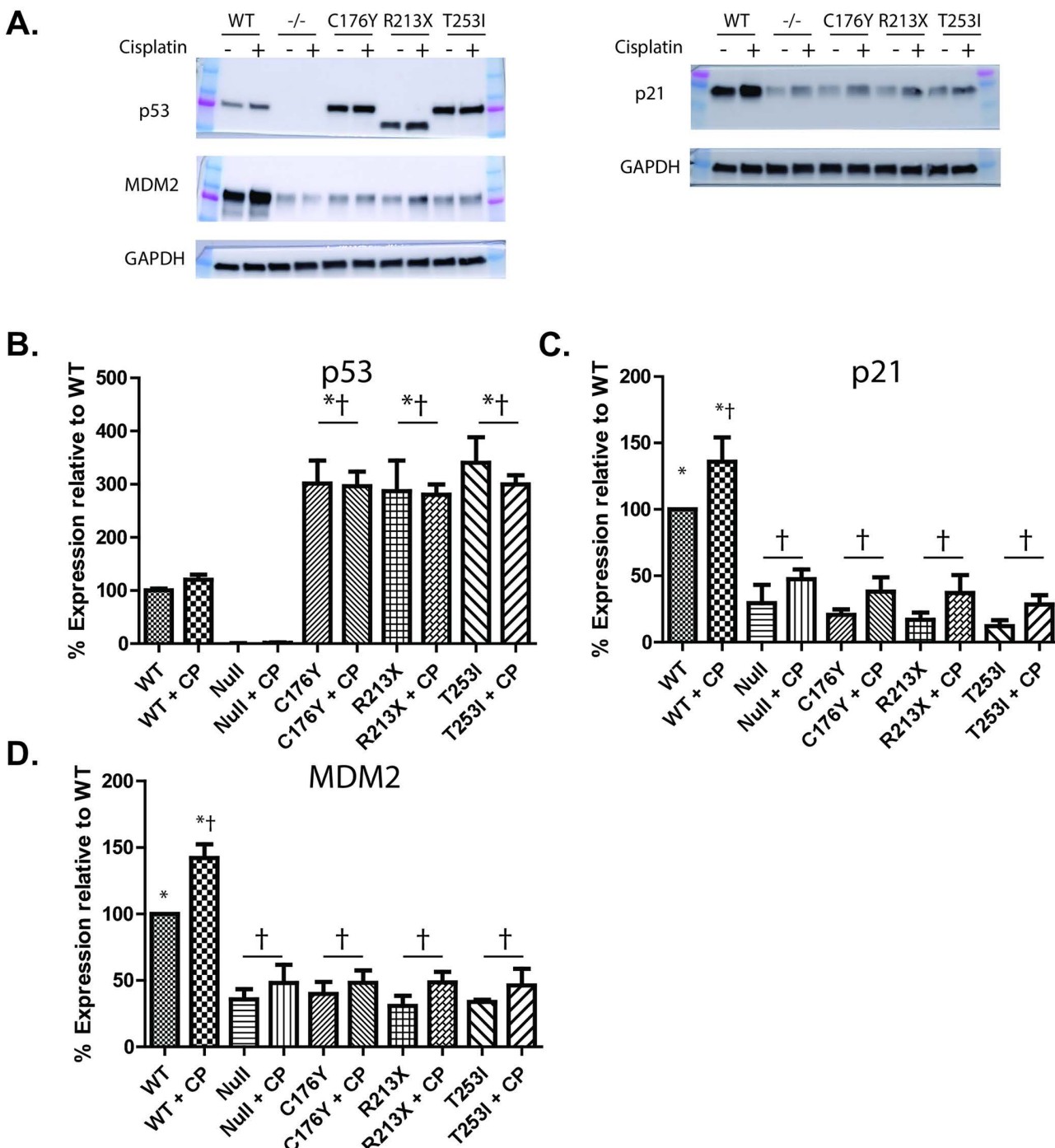

**Fig 3. Impact of p53 T253I on expression of DNA damage responsive proteins.** HEK293 p53[-/-] cells and those complemented with WT-GFP, C176Y-GFP, R213X-GFP, or T253I-GFP p53 were treated with 5 ug/mL Cisplatin (or a DMSO vehicle control) for one hour followed by a 2-hour recovery period. Cells were collected and a Western blot was run looking at p53 signaling events. Levels of p53, MDM2, (**A, left side**) and p21 (**A, right side**) were examined with or without damage in each of the p53 conditions (null, WT, and the three mutants). Graphical depictions of p53 (**B**), p21 (**C**), and MDM2 (**D**) are shown. Data represent the mean expression levels, normalized to GAPDH, of three experimental replicates and significance is denoted by * or † (* indicates $p < 0.05$ when comparing the sample to p53 -/-, † $p < 0.05$ when comparing the sample to p53 WT).

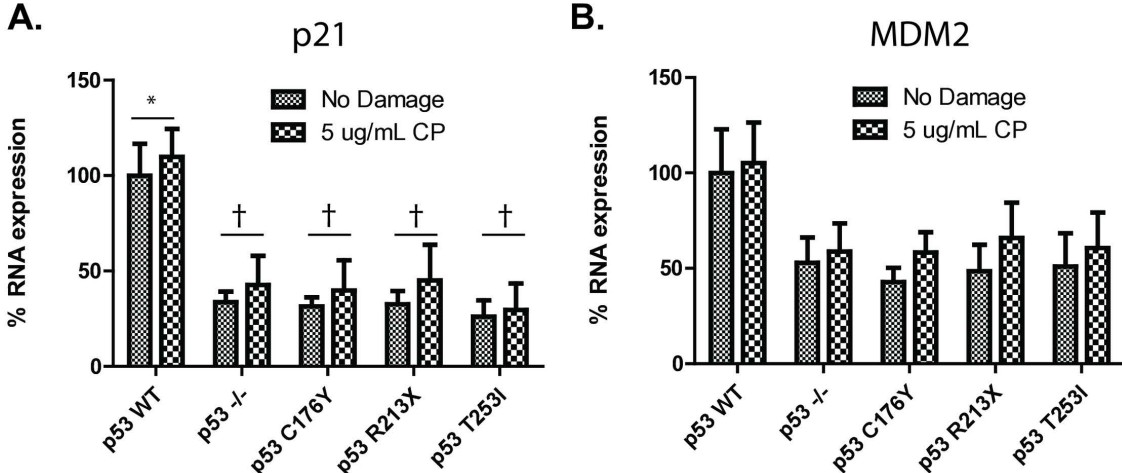

**Fig 4. Impact of p53 T253I on DNA damage responsive mRNA expression.** HEK293 p53[-/-] cells and those complemented with WT-GFP, C176Y-GFP, R213X-GFP, or T253I-GFP p53 were treated with 5 ug/mL Cisplatin (or a DMSO vehicle control) for one hour followed by a 2-hour recovery period. RNA was isolated from these cells and subjected to rtPCR in the presence of primers for p21 (**A**) and MDM2 (**B**). Data represent the mean RNA expression levels of three experimental replicates and significance is denoted by * or † (* indicates $p < 0.05$ when comparing the sample to p53 -/-, † $p < 0.05$ when comparing the sample to p53 WT).

hypothesize that the p53 T253I mutation may be pathogenic and represent a loss-of-function mutation, similar to R213X and C176Y, two other known LFS-associated variants [14,15]. LFS is a cancer predisposition syndrome associated with life-long higher risk of malignancy [3,4], and patient outcomes have been shown to be improved by appropriate cancer surveillance [9,10,32]. Our goal was to establish T253I as an LFS-associated *TP53* mutation to justify personalized cancer surveillance and appropriate genetic counseling for our patient and other patients with the same germline variant.

Loss-of-function mutations in *TP53* may contribute to carcinogenesis through interference with p53-mediated DNA damage responses [33]. Consistent with our observations of T253I both in the cancer tissue and cultured cells, many loss-of-function p53 mutations are known to result in high cellular levels of the mutant p53 protein because of impaired p53-induced MDM2 induction and reduced autoregulatory feedback [34]. Although it has been suggested that elevated mutant p53 levels may function in an oncogenic manner by exerting antimorphic (dominant negative) effects on residual WT p53 [35–39], recent reports have argued against this possibility [40], and our studies have not clarified whether T253I p53 may impact residual WT p53 function. We realize that the cell responses of mutant *TP53* in the background of an otherwise *TP53*-deficient cell (as in our studies) may differ from heterozygous expression as would be expected in germline LFS carriers. However, we reasoned that the p53 null background would enable clarity of the impact of p53 T253I on p53 tumor suppressor function. We acknowledge that our study design did not investigate the impact of each variant studied on cellular damage and survival responses in the heterozygous setting which would model the germline LFS state. Instead, we decided to interrogate the T253I variant to understand its impact on p53 responses in isolation, but we are planning follow-up studies to determine how T253I p53 interacts with wt p53 and its impact on cell responses in a heterozygous setting.

Using the system of p53 complementation into a p53 [-/-] background, we found evidence that p53 T253I exhibits attenuated p53 signaling responses including damage-induced p21 and MDM2 induction and reduced DNA binding and transcriptional activity compared to p53 WT, comparable to established LOF mutants (Figs 3–5). We realize that tagging p53 proteins with GFP may alter their behavior, however we observed that GFP-tagged p53 WT cells complemented the loss of p53 signaling seen in p53 -/- cells, and that GFP-tagged p53 mutant cells exhibit attenuated responses. Since each p53

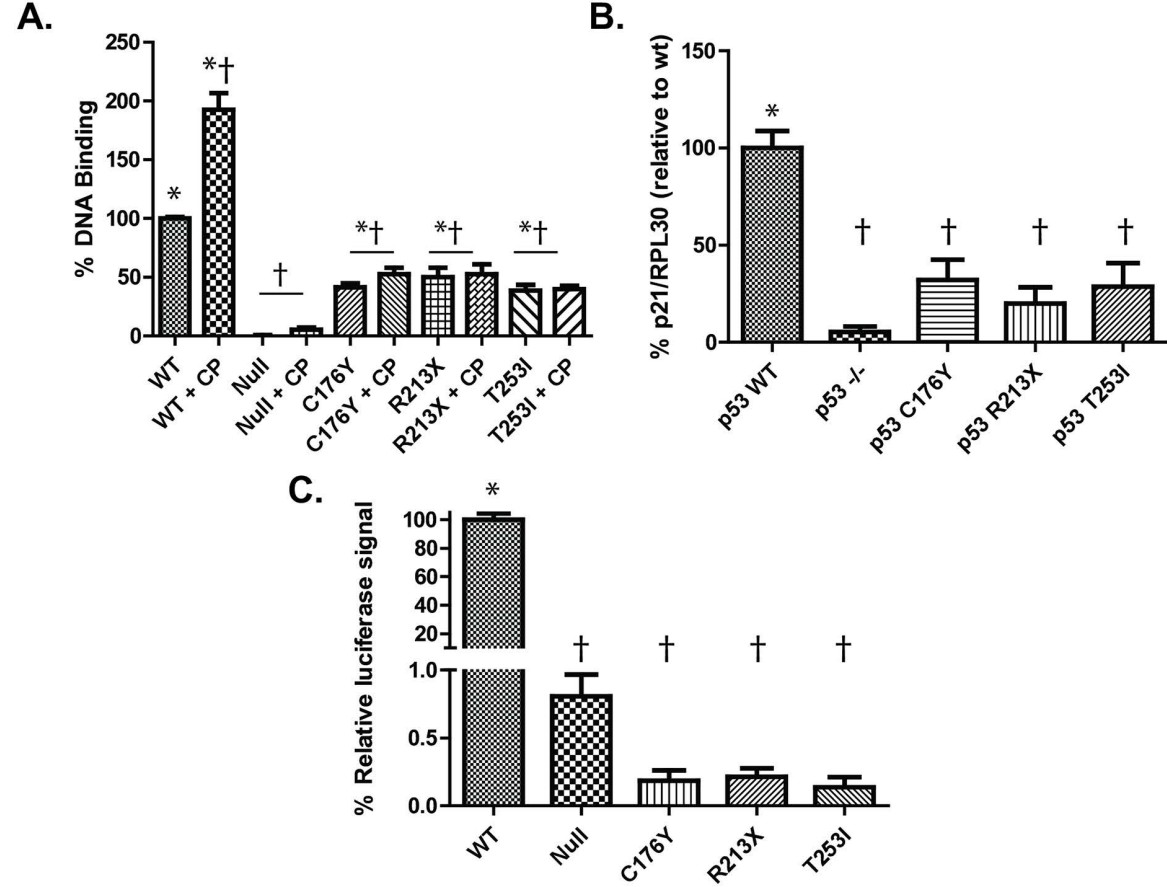

**Fig 5. Mutant p53 proteins have impaired DNA binding and transcriptional regulation.** HEK293 p53⁻/⁻ cells and those complemented with WT-GFP, C176Y-GFP, R213X-GFP, or T253I-GFP p53 were assayed for DNA binding capability and transcriptional regulatory capacity. **(A)** Log phase cells were exposed 5 μg/mL Cisplatin (or a DMSO vehicle control) for 24 hours. Nuclear protein was isolated from cells and a DNA binding assay was performed. Binding efficiency was measured by the ability of nuclear p53 to recognize and bind to consensus p53 binding DNA sequences attached to 96-well plate wells. Data represent the binding strength of each condition relative to undamaged p53 WT cells and are expressed as the mean of three experimental replicates. **(B)** DNA binding capacity of HEK p53⁻/⁻ and p53 ⁻/⁻+WT, C176Y, R231X, or T253I p53 log phase cells was measured using CHIP-PCR. Genomic DNA was isolated, fragmented, and immunprecipitated with p53 antibody. After separating the DNA, primers to p21 (and RPL30 as a control) were used to identify the relative quantity of bound p21 promoter DNA in each p53 condition through PCR. Data represent the mean of three experimental replicates. **(C)** Transcriptional regulatory capacity was measured using a dual-glow luciferase assay. Cells were co-transfected with *luc2P/p53 RE* and *hRluc/CMV* vectors at a 10:1 ratio. After 24 hours the expression of these genes was measured by a stop-and-glow (Promega) system. For each p53 condition, luminescent intensity was calculated as a ratio of *luc2P* to *hRluc* and expressed as a percentage of intensity relative to WT-GFP-complemented p53⁻/⁻ cells. Data are expressed as the mean of three experimental replicates and significance in **(A)** **(B)** and **(C)** is denoted by * or † (* indicates $p < 0.05$ when comparing the sample to p53 -/-, † $p < 0.05$ when comparing the sample to p53 WT).

mutant contained the identical GFP tag as the wt p53, we consider the presence of a GFP tag to be unlikely to account for attenuated responses of any of the p53 mutants and conclude that this issue does not detract from the primary observation in the manuscript (i.e., that T253I p53 exhibits loss of function).

Together with the clinical observation of an LFS-characteristic tumor in a patient with a germline p53 T253I mutation, our studies suggest that p53 T253I results in a loss of p53 function and may establish a pro-carcinogenic cellular state consistent with a cancer predisposition syndrome such as LFS. It is important to point out that while the patient harboring this mutation developed an ACC at a very young age, the patient's father – the only other member of the family confirmed as a carrier of *TP53* c.758C>T (T253I) – has not presented with cancer to date. This raises the possibility that that

germline p53 T253I may be variably penetrant, consistent with prior observations of hypomorphic p53 mutations in ACCs [41], or it could simply be that that father has not developed a cancer yet and remains at high risk of doing so.

As more genetic variants are identified in cancer-relevant genes, it is increasingly important to determine functional significance to clinically observed mutations to guide clinical management as well as to gain insight into disease pathophysiology. Proper classification of the p53 T253I mutation as an LFS-associated pathogenic variant is important to develop a rational cancer-preventive and early surveillance plan to improve clinical outcomes for our patient, their family and for other patients with germline p53 T253I mutations.

## Supporting information

**S1 Table. An itemized list of antibodies used in this manuscript.**
(PDF)

**S1 Fig. Histological staining of tumor tissue section shows classical signs of ACC as well as suggests p53 abnormalities.** (Top) Immunohistochemical staining of the ACC showing that tumor cells were immunoreactive for synaptophysin and Melan-A, with increased Ki67; total tissue Ki67 positivity was estimated at 15%. (Bottom) Neoplastic tissue contained higher levels of total p53 protein than adjacent non-neoplastic tissue.
(PDF)

**S2 Fig. Complementation of HEK p53$^{-/-}$ cells with p53 WT-GFP.** (A) A Western blot of HEK p53$^{+/+}$, HEK293 p53$^{-/-}$, and WT-GFP-complemented HEK p53$^{-/-}$ cells, as well as a graphical depiction (B) of the relative abundances of p53 in each cell line. Data represent the mean expression levels, normalized to GAPDH, of at least three experimental replicates.
(PDF)

**S3 Fig. Location of incorporation of p53 mutants into HEK293 cells.** A cartoon schematic of the functional domains of the p53 protein overlaid with the locations of the mutations being examined in this study. Note that all three mutations lie within the DNA binding domain and that both single nucleotide polymorphisms (C176Y and T253I) are further located within highly conserved regions of the DNA binding domain of p53.
(PDF)

**S4 Fig. Steady state expression levels of p53 and MDMD2 in HEK293 p53$^{-/-}$ cells complemented with WT-GFP, C176Y-GFP, R213X-GFP, or T253I-GFP p53.** A Western blot of HEK p53$^{-/-}$ with or without complementation with WT-GFP, C176Y-GFP, R213X-GFP, or T253I-GFP p53 as well as a graphical depiction of the relative abundances (B) of p53 and MDM2 in each cell line. Data represent the mean expression levels, normalized to GAPDH, of three experimental replicates.
(PDF)

**S5 Fig. Timecourse of recovery after cisplatin treatment shows maximal induction window for p53-dependent DNA damage responses.** (A) WT-GFP-complemented HEK p53$^{-/-}$ cells were treated with 5 µg/mL for one hour (or untreated) and then allowed to recover for a period of time ranging from 1–24 hours. At the end of the recovery period, cells were harvested and a Western Blot was run to measure the expression of p53, MDM2, and p21, using GAPDH as a loading control. (B) A graphical depiction of the relative abundances of p53, MDM2, and p21, normalized to GAPDH, at each timepoint after damage for the duration of the recovery period. Data were presented as the percentage of expression relative to non-damaged (ND) cells and represent the average of three replicates.
(PDF)

**S6 Fig. Luciferase signal strength from *luc2P/p53 RE* and *hRluc/CMV* reporters.** (A) p53-driven firefly luciferase reporter assay and (B) CMV-driven *renilla* reporter assay results that were combined to generate the ratio as seen in

Fig 5b. Data represent the mean luciferase signal, normalized to p53 WT-GFP expressing cells, of three experimental replicates. Samples marked with an * are not significantly different from each other.
(PDF)

## Acknowledgments

We are grateful to the Biostatistics and Bioinformatics, Flow Cytometry and Immune Monitoring, Biospecimen Procurement and Translational Pathology, Cancer Research and Informatics, and Oncogenomics Shared Resource Facilities of the University of Kentucky Markey Cancer Center for their technical and scientific contributions to the work. We thank the COBRE Imaging Core facility at the University of Kentucky for providing service with microscopes and imaging systems. We are indebted to the Clinical Research Office of the Division of Pediatric Hematology/Oncology at the University of Kentucky for their efforts in accrual and data management for Project Inherited Cancer Risk, the clinical study to identify and manage children, adolescents and young adults with cancer predisposition syndromes. Finally, we are immensely grateful to the patients and families who consented to take part in the "Project Inherited Cancer Risk" study at the University of Kentucky by which these variants were identified.

## Author contributions

**Conceptualization:** Nathaniel C. Holcomb, Amanda M. Harrington, John A. D'Orazio.

**Data curation:** Nathaniel C. Holcomb, Hong Pu, Berina Halilovic, Shulin Zhang, Catherine Sears.

**Formal analysis:** Nathaniel C. Holcomb, Amanda M. Harrington, Hong Pu, Berina Halilovic, Nathan R. Shelman, Shulin Zhang, Terra Armstrong, Brent Shelton, Lauren Corum, John A. D'Orazio.

**Funding acquisition:** John A. D'Orazio.

**Investigation:** Nathaniel C. Holcomb, Hong Pu, Berina Halilovic, Shulin Zhang, Catherine Sears, John A. D'Orazio.

**Methodology:** Nathaniel C. Holcomb, Hong Pu, Berina Halilovic, Nathan R. Shelman, Shulin Zhang, Catherine Sears.

**Project administration:** John A. D'Orazio.

**Resources:** Nathaniel C. Holcomb, Terra Armstrong, John A. D'Orazio.

**Supervision:** Nathaniel C. Holcomb, John A. D'Orazio.

**Validation:** Nathaniel C. Holcomb, Hong Pu, Berina Halilovic, Shulin Zhang, Catherine Sears.

**Writing – original draft:** Nathaniel C. Holcomb, Amanda M. Harrington, John A. D'Orazio.

**Writing – review & editing:** Nathaniel C. Holcomb, Amanda M. Harrington, Nathan R. Shelman, Brent Shelton, John A. D'Orazio.

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
