## [Decision Letter · Decision Letter 0]

2 May 2025

Dear Dr. D'Orazio,

Thank you for submitting your manuscript to PLOS ONE. After careful consideration, we feel that it has merit but does not fully meet PLOS ONE’s publication criteria as it currently stands. Therefore, we invite you to submit a revised version of the manuscript that addresses the points raised during the review process.

We look forward to receiving your revised manuscript.

Kind regards,

Swati Palit Deb

Academic Editor

PLOS ONE

**Journal Requirements:**

1. When submitting your revision, we need you to address these additional requirements. Please ensure that your manuscript meets PLOS ONE's style requirements, including those for file naming. The PLOS ONE style templates can be found at https://journals.plos.org/plosone/s/file?id=wjVg/PLOSOne_formatting_sample_main_body.pdf and https://journals.plos.org/plosone/s/file?id=ba62/PLOSOne_formatting_sample_title_authors_affiliations.pdf 2. Thank you for stating the following financial disclosure: Kentucky Pediatric Cancer Research Trust FundNCI P30 CA177558NIH P20 GM121327  Please state what role the funders took in the study.  If the funders had no role, please state: "The funders had no role in study design, data collection and analysis, decision to publish, or preparation of the manuscript." If this statement is not correct you must amend it as needed. Please include this amended Role of Funder statement in your cover letter; we will change the online submission form on your behalf. 3. Thank you for stating the following in the Acknowledgments Section of your manuscript: This research was supported by the Biostatistics and Bioinformatics, Flow Cytometry and Immune Monitoring, Biospecimen Procurement and Translational Pathology, Cancer Research and Informatics, and Oncogenomics Shared Resource Facilities of the University of Kentucky Markey Cancer Center, each supported by the University of Kentucky Markey Cancer Center’s Cancer Center Support Grant (P30 CA177558).  Additional support was provided by the Kentucky Pediatric Cancer Research Trust Fund, the Joy Wills Endowment for Childhood Cancer Research, and the DanceBlue Philanthropic Organization.  We thank the COBRE Imaging Core facility at the University of Kentucky supported by P20 GM121327 for providing service with microscopes and imaging systems.  We are indebted to the Clinical Research Office of the Division of Pediatric Hematology/Oncology at the University of Kentucky for their efforts in accrual and data management for Project Inherited Cancer Risk, a clinical study to identify and manage children, adolescents and young adults with cancer predisposition syndromes. We note that you have provided funding information that is not currently declared in your Funding Statement. However, funding information should not appear in the Acknowledgments section or other areas of your manuscript. We will only publish funding information present in the Funding Statement section of the online submission form. Please remove any funding-related text from the manuscript and let us know how you would like to update your Funding Statement. Currently, your Funding Statement reads as follows: Kentucky Pediatric Cancer Research Trust FundNCI P30 CA177558NIH P20 GM121327   Please include your amended statements within your cover letter; we will change the online submission form on your behalf. 4. When completing the data availability statement of the submission form, you indicated that you will make your data available on acceptance. We strongly recommend all authors decide on a data sharing plan before acceptance, as the process can be lengthy and hold up publication timelines. Please note that, though access restrictions are acceptable now, your entire data will need to be made freely accessible if your manuscript is accepted for publication. This policy applies to all data except where public deposition would breach compliance with the protocol approved by your research ethics board. If you are unable to adhere to our open data policy, please kindly revise your statement to explain your reasoning and we will seek the editor's input on an exemption. Please be assured that, once you have provided your new statement, the assessment of your exemption will not hold up the peer review process. 5. Please review your reference list to ensure that it is complete and correct. If you have cited papers that have been retracted, please include the rationale for doing so in the manuscript text, or remove these references and replace them with relevant current references. Any changes to the reference list should be mentioned in the rebuttal letter that accompanies your revised manuscript. If you need to cite a retracted article, indicate the article’s retracted status in the References list and also include a citation and full reference for the retraction notice.

Reviewers' comments:

**Comments to the Author**

1. Is the manuscript technically sound, and do the data support the conclusions?

Reviewer #1: Yes

Reviewer #2: Yes

2. Has the statistical analysis been performed appropriately and rigorously?

Reviewer #1: No

Reviewer #2: No

3. Have the authors made all data underlying the findings in their manuscript fully available?

Reviewer #1: Yes

Reviewer #2: Yes

4. Is the manuscript presented in an intelligible fashion and written in standard English?

Reviewer #1: Yes

Reviewer #2: Yes

**Reviewer #1: ** General Comments -

This study presents a laudable, detailed, functional analysis of a TP53 mutation hypothesized to be pathological for Li-Fraumeni Syndrome (LFS). The data demonstrates that the mutant in question behaves almost identically to 2 other known pathogenic mutants in the test assays utilized.

Specific Recommendations _

1. The design of this study does not exactly replicate the genetics of LFS, in which the mutant TP53 allele is heterozygous with a wild-type allele. This raises the question as to whether any of the 3 mutant alleles studied here may behave differently in these assays if their wild-type partner was present?

2. Since the 2 known pathogenic TP53 alleles are such a critical component of this analysis, it would be preferable for them to appear in the Introduction, along with appropriate referencing. The reference provided where they're currently introduced in the Discussion (Zhang et al., 2014) appears to only support the pathogenicity of R213X, not C176Y.

3. There is a major statistical issue for most of the bar charts, apparently with designation of significance. It appears that whatever symbols supposed to appear were substituted by undefined a-d lettering. In addition, once these designations are corrected, the p-values they're associated with and what bars they reference needs to be defined in every Figure Legend where they're shown.

4. In Fig. 1B, the Null & WT bars should be swapped to match order of all other Figs.

5. In Fig. S-2B, the 'Parental p53' designation is confusing & vague. Perhaps just 'WT p53' or 'p53 +/+' might serve better?

**Reviewer #2:**  1. It would be important to show whether there is any difference in function between GFP-tagged vs untagged P53?

2. Although authors stated that p53 T253I has less transcriptional activity due to inefficient DNA binding through an ELISA-based assay, it would be more convincing to show with ChIP-qPCR for a couple of known p53-dependent regulatory genes whether such observation still holds for a complementary assay system.

**Do you want your identity to be public for this peer review?** For information about this choice, including consent withdrawal, please see our Privacy Policy

Reviewer #1: No

Reviewer #2: No

---

## [Author Response · Author response to Decision Letter 1]

25 Sep 2025

please refer to the attached "Response to Reviewers" file. thank you.

---

## [Editor Report · Decision Letter 1]

30 Oct 2025

Characterization of p53 p.T253I as a pathogenic mutation underlying Li-Fraumeni Syndrome

PONE-D-25-07425R1

Dear Dr. John August D'Orazio

We’re pleased to inform you that your manuscript has been judged scientifically suitable for publication and will be formally accepted for publication once it meets all outstanding technical requirements.

Kind regards,

Swati Palit Deb

Academic Editor

PLOS ONE
---

## [Editor Report · Acceptance letter]

PONE-D-25-07425R1

PLOS ONE

Dear Dr. D'Orazio,

I'm pleased to inform you that your manuscript has been deemed suitable for publication in PLOS ONE. Congratulations! Your manuscript is now being handed over to our production team.

Kind regards,

on behalf of

Dr. Swati Palit Deb

Academic Editor

PLOS ONE